# Heme Oxygenase-1 Inhibition Sensitizes Human Prostate Cancer Cells towards Glucose Deprivation and Metformin-Mediated Cell Death

**DOI:** 10.3390/ijms20102593

**Published:** 2019-05-27

**Authors:** Marco Raffaele, Valeria Pittalà, Veronica Zingales, Ignazio Barbagallo, Loredana Salerno, Giovanni Li Volti, Giuseppe Romeo, Giuseppe Carota, Valeria Sorrenti, Luca Vanella

**Affiliations:** 1Department of Drug Science, Biochemistry Section, University of Catania, 95125 Catania, Italy; marco.raffaele@hotmail.com (M.R.); veronica_zingales@libero.it (V.Z.); ignazio.barbagallo@unict.it (I.B.); giuseppe-carota@outlook.it (G.C.); sorrenti@unict.it (V.S.); 2Department of Drug Science, Pharmaceutical Chemistry Section, University of Catania, 95125 Catania, Italy; vpittala@unict.it (V.P.); l.salerno@unict.it (L.S.); gromeo@unict.it (G.R.); 3Department of Biomedical and Biotechnological Sciences, University of Catania, 95125 Catania, Italy; livolti@unict.it

**Keywords:** prostate cancer, heme oxygenase, metformin, apoptosis, ER stress, HO-1 activity inhibitor

## Abstract

High levels of heme oxygenase (HO)-1 have been frequently reported in different human cancers, playing a major role in drug resistance and regulation of cancer cell redox homeostasis. Metformin (MET), a drug widely used for type 2 diabetes, has recently gained interest for treating several cancers. Recent studies indicated that the anti-proliferative effects of metformin in cancer cells are highly dependent on glucose concentration. The present work was directed to determine whether use of a specific inhibitor of HO-1 activity, alone or in combination with metformin, affected metastatic prostate cancer cell viability under different concentrations of glucose. MTT assay and the xCELLigence system were used to evaluate cell viability and cell proliferation in DU145 human prostate cancer cells. Cell apoptosis and reactive oxygen species were analyzed by flow cytometry. The activity of HO-1 was inhibited using a selective imidazole-based inhibitor; genes associated with antioxidant systems and cell death were evaluated by qRT-PCR. Our study demonstrates that metformin suppressed prostate cancer growth in vitro and increased oxidative stress. Disrupting the antioxidant HO-1 activity, especially under low glucose concentration, could be an attractive approach to potentiate metformin antineoplastic effects and could provide a biochemical basis for developing HO-1-targeting drugs against solid tumors.

## 1. Introduction

Heme oxygenase-1 (HO-1) is the inducible isoform of heme oxygenase, the first rate-limiting enzyme in the degradation of heme to free iron, carbon monoxide (CO), and biliverdin [1].

HO-1 is present at low levels in many tissues and is highly upregulated by numerous stimuli, such as heme, heavy metals, UV irradiation, reactive oxygen species (ROS), polyphenols, and inflammatory cytokines [2,3]. HO-1 is mainly localized in microsomes, but it has also been demonstrated to be differently localized in caveolae, mitochondria, and nuclei [4,5,6].

Endogenous induction of HO-1 is widely acknowledged as an adaptive cellular response, able to counteract oxidative stress. Moreover, HO-1-derived metabolites have several protective effects on cells and tissues against injuries related to pathological conditions like diabetes, obesity, and cardiovascular diseases [7,8,9,10,11,12,13].

High levels of HO-1 have been frequently reported in different human cancers [14,15,16], playing a major role in drug resistance and regulation of cancer cell redox homeostasis [17,18,19,20,21,22].

Elevated HO-1 levels have been shown in many cancers, as reported by Jozkowicz et al. [23]. A previous study by Florczyk et al. [24] revealed that enhanced activity of biliverdin reductase may protect cells in stressful conditions arising from anti-cancer drugs, cisplatin, and doxorubicin. Additionally, data from Banerjee et al. [25] demonstrate that HO-1 is up-regulated in renal cancer cells as a survival strategy against chemotherapeutic drugs and promotes growth of tumor cells by inhibiting both apoptosis and autophagy. Thus, application of chemotherapeutic drugs along with HO-1 inhibitor may elevate therapeutic efficiency by reducing the cytoprotective effects of HO-1 and by simultaneous induction of both apoptosis and autophagy.

Metformin (MET), a drug widely used for type 2 diabetes, has recently gained interest for treating several cancers [26]. The anti-proliferative effects of metformin, reported in several cancers including breast, colon, glioma, ovarian, pancreatic, and prostate cancer [27,28,29], have been mainly associated with the capacity of MET to inhibit mitochondrial respiration and consequently increasing glycolysis rates [30] and to arrest cell cycle and inducing caspase-dependent apoptosis [31,32].

An inverse relationship has been found between the progress of prostate cancer and type 2 diabetes in patients who use metformin.

Additionally, MET use had a trend of improving survival for prostate cancer patients [33,34]. Metformin was reported to reduce prostate cancer growth prominently under a high fat diet, acting through the modulation of several tumoral-associated processes in a xenograft model of human cell lines, using immunodeficient mice [35].

MET treatment and caloric restriction increase the AMP/ATP ratio and activate AMP-activated protein kinase (AMPK), switching cells from an anabolic to a catabolic state. Treatment of breast cancer cells with MET significantly decreased cholesterol content with concomitant inhibition of various cholesterol regulatory genes [36], suggesting that drugs affecting cholesterol synthesis, such as simvastatin/atorvastatin, could be used in the treatment of cancer [37].

Recent studies showed, in vitro, that MET, in combination with simvastatin, induced G1-phase cell cycle arrest [38], while, in vivo, treatment of mice with a combination of metformin and atorvastatin caused stronger inhibition than either drug used individually on the growth of PC-3 tumors [39]. Furthermore, it has been reported that combination treatment of MET with valproic acid was more effective at slowing prostate tumor growth in vivo compared to either drug alone, in mouse xenografts [40].

MET has been proposed to operate as an agonist of Sirtuin-1, a nicotinamide adenine dinucleotide (NAD+)-dependent deacetylase that mimics most of the metabolic responses to calorie restriction [41].

Most primary and metastatic human cancers show significantly increased glucose uptake because of their enhanced glucose consumption. Cancer cells are more dependent on glucose for energy production than normal cells [42,43]. Recent studies have indicated that the anti-proliferative effects of metformin in cancer cells are highly dependent on the glucose concentration [44,45].

Furthermore, it has been shown that metformin significantly decreases the intracellular glutathione levels and enhanced sensitivity of esophageal squamous cell carcinoma to cisplatin [46], suggesting that regulation of the antioxidant defenses represents a key target for cancer therapy. In this regard, new evidence has shown the involvement of TIGAR (TP53-induced glycolysis and apoptosis regulator) in glutathione restoration [47].

A growing body of independent evidence supports the association between metabolic alterations and the development and progression of prostate cancer, as well as the promising role of MET in controlling prostate cancer outcomes.

The present work was directed to determine whether use of a specific inhibitor of HO-1 activity, alone or in combination with MET, would affect metastatic prostate cancer cell viability under different concentrations of glucose.

## 2. Results

### 2.1. Effect of Metformin on Cell Viability

It has been shown that MET is selectively toxic to p53-deficient cells and provides a potential mechanism for the reduced incidence of tumors observed in patients being treated with metformin [48]. Results obtained by Gonnissen et al. showed that p53-mutant cells were more resistant to MET than PC3 and 22Rv1 [49].

In order to determine whether MET affects proliferation of human p53-mutant prostate cancer cells, we analyzed the effect of the drug on DU145, a p53-mutant cell line [50]. DU145 was treated with different concentrations of MET (3–50 mM) for 24 h, then the cell viability was assessed by MTT assay (Figure 1). In the presence of the highest concentration of MET (50 mM), the cell viability was reduced by about 50% compared to treatment with the other concentrations, and about 80% compared to untreated cells. Consistent with previous studies, cell treatment with metformin showed significant cytotoxicity. The concentration of 10 mM was used in the following experiments.

### 2.2. Real-Time Analysis of Cell Proliferation in Presence of Metformin and Different Glucose Concentrations

In order to study the effect of MET on DU145 cells proliferation in conditions of glucose deprivation, dynamic changes in cell index were monitored using the xCELLigence system upon exposure to 1 mM glucose (G1 control (CTRL)) or 25 mM glucose (G25 CTRL) for 48 h. The DU145 cells were untreated and treated with 10 mM MET. The cell index of both control groups of untreated cells followed the same trend across the 48 h. After 24 h, the cell index of cell groups treated with MET followed different trends, showing a noticeable gap at the end of 48 h.

As shown in Figure 2, low glucose concentration enhanced MET cytotoxicity in DU145 cells at 24 h and 48 h.

### 2.3. Effect of Metformin on HO-1, CHOP, BAX, and Sirtuins mRNA Expression

The effect of MET on HO-1, CHOP and BAX, genes related to the endoplasmic reticulum stress and apoptosis activation, was assessed by measuring mRNA levels. Their gene expression followed the same trend, with an increased level after MET administration, compared to the control group (Figure 3A–C). In order to analyze the effect of metformin on the pathway related to apoptosis regulation, mRNA levels of different sirtuins were assessed. Treatment with 10 mM MET led to an increase of Sirt1 levels, more pronounced in the highest concentration of glucose. Conversely, Sirt3 and Sirt5 levels were reduced after treatment, in both concentrations of glucose. The low concentration of glucose caused a significant decrease of Sirt3 and Sirt5 levels, even in the absence of MET (Figure 3D–F).

### 2.4. Metformin Enhances the Apoptosis Rate of DU145 in the Presence of a Selective HO-1 Activity Inhibitor

The apoptosis of DU145 cells was measured using Annexin V staining and flow cytometry analysis after 12 h of treatment. As shown in Figure 4, in both concentrations of glucose, the rate of apoptotic DU145 cells was significantly increased after treatment with MET. The co-treatment with a selective inhibitor of HO-1 activity (VP1347) caused a strong enhancement of apoptosis levels. The treatment with VP1347 alone did not demonstrate significant differences to the untreated control. The decreased level of live cells was more evident in the co-treatment group, indicating a synergistic effect (G1 CDI = 0.90; G25 CDI = 0.95) between MET and VP1347.

### 2.5. Effect of Metformin on Oxidative Stress Regulation Pathway

At the 1 mM glucose concentration, MET treatment caused a significant decrease of glutathione (GSH) levels compared to the control, especially when combined with VP1347. The treatment with VP1347 alone was not able to reduce the GSH levels. In the presence of the highest concentration of glucose, all treatments showed a reduction of GSH levels without difference within the groups (Figure 5A). TIGAR and Gamma-Glutamylcysteine Synthetase (GCLC) mRNA expression was measured after 6 h of treatment with MET and VP1347 in both concentrations of glucose (Figure 5B,C). At the 1 mM glucose concentration, TIGAR levels were significantly reduced when treated with MET and VP1347, used alone or in combination, compared to the control. GCLC mRNA expression showed a marked increase only in the group treated with metformin. At high glucose concentration, TIGAR levels were strongly reduced in the presence of MET and VP1347, particularly when used as co-treatment. GCLC levels were decreased in all treated groups compared to the control.

### 2.6. Effect of Metformin and HO-1 Activity Inhibitor on ROS Production

The quantitative measurement of cells undergoing oxidative stress was evaluated by cytometry, using the Muse Oxidative Stress Kit after a 6 h treatment with MET and VP1347. As shown in Figure 6, in both concentrations of glucose, a positive ROS level was notably increased after MET and VP1347 administration compared to the control. The decrease of ROS M1 values was more evident in the co-treatment group, indicating a synergistic effect (G1 CDI = 0.92; G25 CDI = 0.86) between MET and VP1347.

## 3. Discussion

In diabetic patients, metformin decreases plasma glucose concentration mainly by lowering hepatic gluconeogenesis and glucose output. This effect is followed by an increase in glucose uptake and the amelioration of insulin resistance [51]. MET works by targeting the enzyme AMPK (AMP activated protein kinase), which induces muscles to take up glucose from the blood. A recent breakthrough has found the upstream regulator of AMPK to be a protein kinase known as LKB1, a well-recognized tumor suppressor [52].

AMP-activated protein kinase (AMPK) activators have been in use for many years to treat type 2 diabetes, but recent data demonstrate that these compounds can inhibit AKT-derived pro-survival effects and induce apoptosis in cancer cells [53].

Several data suggest that MET, an AMPK inducer, could protect from cancer and inhibit breast and glial tumor cell proliferation [52,54,55] through inhibition of the mitochondrial complex I activity and cellular respiration. An essential role of the electron transport chain in cell proliferation has been reported, due to its ability to enable the biosynthesis of aspartate, a proteinogenic amino acid and a precursor in purine and pyrimidine synthesis [56].

In this study, we show that MET not only is a very potent inhibitor of human prostate cancer cell growth, but its effect, in vitro, is potentiated by low glucose treatment and HO-1 activity inhibition.

Although many studies have shown the antineoplastic properties of MET, the mechanisms of action have not been clearly defined yet. As an anticancer agent, MET weakly induces cancer cell apoptosis. However, as shown in Figure 2, under cell culture conditions with low glucose, MET decreases cell proliferation. Starvation from glutamine or glucose for short periods resulted in cell cycle arrest and apoptosis induction by means of reactive oxygen species generation and mitochondrial dysfunction [57,58].

Glucose deprivation has been previously shown to increase MET-induced cell death in cancer cells [59,60].

Glucose deprivation, a cell condition that occurs in solid tumors, activates the unfolded protein response (UPR), which allows the cell to survive under stress conditions [61].

Cancer cells in solid tumors are not supplied sufficient glucose because they are often distant from blood vessels. Adaptive response mechanisms are required for cancer cells to survive in the tumor microenvironment. The UPR in cancer cells plays an important role in their survival and results in tumor malignancies and antitumor drug resistance [62]. If cancer cells have no adaptive response, such as the UPR, activated by endoplasmic reticulum (ER) stress, they would not be able to escape death under glucose deprivation conditions [63].

The acute increase of *CHOP* expression leads to activation of the mitochondria-mediated apoptosis pathway [64]. Therefore, ER stress-dependent apoptosis has been recently reported as a promising therapeutic pathway to target for inducing cancer cell death [65].

Previous studies have shown that metformin induces ER stress and UPR-related genes and inhibited cancer cell proliferation in a dose-dependent manner [66,67,68]. Aside from IRE1a, PERK, ATF6a, and CHOP regulation, ER stress has been associated with HO-1 induction, which represents a cytoprotective adaptive response to survive stringent conditions [69].

Previously, it has also been reported that the high expression of HO-1 is associated with tumor invasiveness and poor clinical outcome in non-small cell lung cancer patients [16].

Our previous studies showed that the chemotherapeutic drugs, carfilzomib and bortezomib, both increased HO-1 protein and gene expression via the activation of the UPR response, triggered by ER stress [17,70]. As shown in Figure 3, although glucose deprivation already increased HO-1 levels, MET significantly further induced HO-1, CHOP, and BAX mRNA levels, suggesting metformin can induce ER stress and cell apoptosis.

Sirtuins are homologs of the yeast *SIR2* gene, and their function as regulators in a wide range of biological processes is mostly associated with a nicotinamide adenine dinucleotide (NAD+)-dependent deacetylation [71]. Recent studies have shown that seven sirtuins (SIRT 1–7) are linked to tumor growth regulation. Tumor suppression is associated with upregulation of SIRT1 levels, whereas SIRT3 and SIRT5 act differently as tumor promoters [72,73,74]. DNA damage, exerted by anti-proliferative drugs and the consequent increase of oxidative stress, could induce the expression of transcription factor E2F1, which increases the transcription of SIRT1 and other several apoptotic proteins [75]. Our results showed that MET treatment, both in the presence of 25 mM and 1 mM glucose concentration, increased levels of SIRT1 mRNA, as a demonstration of its anti-proliferative effect [76,77,78]. SIRT3 and SIRT5 are localized to mitochondria and, as previously published, metformin primarily acts through the mitochondrial respiratory chain inhibition [79,80]. Dysfunctional mitochondria probably represent the main cause of the observed decrease of sirtuins 3 and 5 mRNA in prostate cancer cells [71,81,82,83].

In this study, we evaluated MET-induced apoptosis in prostate cancer cells, using Annexin cytometry analysis. As shown in Figure 4, MET induced apoptosis in prostate cancer cells and, consistently with previous published results [17,84], HO-1 activity inhibition was able to further increase the cytotoxic effect induced by metformin in DU145 cells, indicating a synergistic effect between the two drugs.

HO-1 represents the inducible isoform of one of the main cytoprotective systems against oxidative stress and inflammation [23,85,86]. When normal cells are exposed to stress conditions, HO-1 induction is the physiological response in order to guarantee regulation of redox homeostasis. For this reason, an adequate expression of this protein is necessary to confer a basal protection against oxidative stress [87,88]. Cancer cells are often characterized by an overexpression of HO-1 and an enhancement of the cytoprotection systems [16,89,90,91]. It has been reported that metformin alters cellular responses to oxidative stress [92], and the direct activity on mitochondria leads to ROS promotion [30,93,94,95].

The combined treatment with MET and a selective HO-1 activity inhibitor, under cell culture conditions with low and high glucose, demonstrated a significant decrease of GSH levels, TIGAR, and GCLC mRNA levels. Several reports showed the key role of these factors in the oxidative stress management [96,97]. TIGAR is a protein involved in the switch from glycolysis to the pentose phosphate pathway that promotes the production of cellular nicotinamide adenine dinucleotide phosphate (NADPH) [47,98,99]. NADPH levels enhancement, caused by TIGAR upregulation, leads to an increased restoration of GSH [100]. On the contrary, it has been shown that TIGAR knockdown decreased GSH and NADPH production and increased the levels of ROS [101]. In order to confirm the loss of the physiological protection from oxidative stress, ensured by GSH, we measured the ROS level in the presence of MET and the selective HO-1 activity inhibitor. We observed a considerable increase of ROS levels after treatment with the combination of MET and the HO-1 activity inhibitor. These results suggest that HO-1 inhibition may enhance metformin cytotoxicity through triggering ER stress-associated apoptosis and that ROS is also involved in the activation of ER stress, as schematized in the Figure 7.

In conclusion, our study demonstrates that metformin suppressed prostate cancer growth in vitro and increased oxidative stress under low and high glucose conditions.

Our findings show that disrupting the antioxidant HO-1 activity, especially under low glucose concentration, could be an attractive approach to potentiate MET antineoplastic effects and could provide a biochemical basis for developing HO-1-targeting drugs against solid tumors.

Further investigation of metformin’s molecular mechanism and targets will reveal its potential application as a monotherapy or part of a polytherapy associated with a clinically approved HO-1 inhibitor in cancer treatment.

## 4. Materials and Methods

### 4.1. Cell Culture

The human androgen-independent prostate cancer cell line DU145 was cultured in Dulbecco’s modified Eagle’s medium (DMEM) 4.5 g/L glucose, supplemented with 10% FBS, 1% penicillin, and streptomycin at 37 °C and 5% CO_2_. The cells were purchased from American Type Culture Collection (Rockville, MD, USA).

### 4.2. Cell Viability Assay

DU145 cells were seeded at a concentration of 2 × 10^5^ cells per well of a 96-well, flat-bottomed 200 μL microplate. Cells were incubated at 37 °C in a 5% CO_2_ humidified atmosphere and maintained in the presence and absence of different concentrations (3, 5, 10, and 50 mM) of metformin (Sigma Chemical Co., St Louis, MO, USA) for 24 h. Three hours before the end of the treatment time, 20 μL of 0.5% 3-(4,5-dimethylthiazol-2-yl)-2,5-diphenyltetrazolium bromide (MTT) in phosphate-buffered saline (PBS) was added to each microwell. After incubation with the reagent, the supernatant was removed and replaced with 100 μL DMSO to dissolve the formazan crystals produced. The amount of formazan is proportional to the number of viable cells present. The optical density was measured using a microplate spectrophotometer reader (Synergy HT, BioTek) at *λ* = 570 nm.

### 4.3. Real-Time Monitoring of Proliferation

Real-time monitoring of cell proliferation was performed using the xCELLigence RTCA DP system [102]. E-plate 16, used with the xCELLigence system, is a single-use 16-well cell culture plate with bottom surfaces covered with microelectrode sensors (0.2 cm^2^ well surface area; 243 ± 5 µL maximum volume). Real-time changes in electrical impedance were measured using the gold microelectrodes and expressed as cell index”, defined as (Rn-Rb)/15, where Rb is the background impedance and Rn is the impedance of the well with cells. Negative control groups (wells containing 200 µL culture medium without cells with cell index values around 0) were tested in every experiment; however, they were not shown in figures in order to simplify the representations. Before seeding cells into E-plate 16, the background impedance was measured after the addition of 100 µL medium and a 30 min incubation period at room temperature. Following the seeding of the appropriate number of cells into the wells, the plate incubated at room temperature for 30 min in order to allow cell settling. Cell proliferation was monitored every 30 min for 48 h.

### 4.4. Oxidative Stress Assay

The quantitative measurement of cellular populations undergoing oxidative stress was performed using the Muse Oxidative Stress Kit (Merck Millipore, Billerica, MA, USA), according to the manufacturer’s instructions. This assay utilizes dihydroethidium (DHE), which is cell membrane-permeable and, upon reaction with superoxide anions, undergoes oxidation to form DNA-binding fluorophore. The kit determines the percentage of cells that are negative [ROS(−)] and positive [ROS(+)] for reactive oxygen species. Briefly, after 6 h of treatment, 1 × 10^6^ cells/mL were harvested, washed with PBS, and then incubated in the dark at 37 °C for 30 min with the Muse Oxidative Stress Reagent working solution, which contained DHE. The count and percentage of cells undergoing oxidative stress were quantified using the Muse Cell Analyzer and Muse analysis software (Merck Millipore, USA).

### 4.5. Annexin V Assay

Apoptosis determination by Annexin V staining was carried out using Muse Cell Analyzer with the kit provided by the manufacturer Merck Millipore. In brief, 1 × 10^6^ cells were seeded in six-well plates and, after overnight adherence, they were treated and incubated with 10 mM MET and VP1347, 1-{4-[(4-bromobenzyl)oxy]phenyl}-2-(1H-imidazol-1-yl)ethanol, a HO-1 activity non-competitive inhibitor, that binds to the complex enzyme/substrate. The compound was synthesized by Salerno et al. from the University of Catania as previously described [103].

It presented an IC_50_ < 1 µM on HO-1 and showed a >100 selectivity toward HO-2. After 12 h, the cells were detached by trypsinization, centrifuged, and resuspended in PBS. Cell suspension (100 µL) was added with 100 µL of Annexin V reagent and incubated for 20 min at room temperature, after which the cells were analyzed for apoptosis.

### 4.6. RSH Evaluation

For total thiol groups (RSH) determination, DU145 cells were cultured for 24 h in the presence or absence of 10 mM MET and VP1347. Determination of RSH was performed as previously described [104]. In short, this spectrophotometric assay is based on the reaction of thiol groups with 2,2-dithio-bis-nitrobenzoic acid (DTNB) in absolute ethanol to give a colored compound absorbing at λ = 412 nm. We then carried out the removal of proteins with an excess of absolute ethanol, followed by centrifugation at 3000× *g* for 10 min at room temperature. Each value represents the mean ± SD of three experimental determinations for each sample. Results were expressed as nanomoles per milligram of protein.

### 4.7. RNA Extraction and qRT-PCR

RNA was extracted by Trizol reagent (Invitrogen, Carlsbad, CA, USA). First strand cDNA was then synthesized with Applied Biosystem (Foster City, CA, USA) reverse transcription reagent. Quantitative real-time PCR was performed in Step One Fast Real-Time PCR System Applied Biosystems, using the SYBR Green PCR MasterMix (Life Technologies, Monza, Italy). The specific PCR products were detected by the fluorescence of SYBR Green, the double stranded DNA binding dye. The relative mRNA expression level was calculated by the threshold cycle (Ct) value of each PCR product and normalized with that of actin by using a comparative 2^−ΔΔ*C*t^ method.

### 4.8. Statistical Analysis

Statistical significance (*p* < 0.05) of differences between experimental groups was determined by the Fisher method for analysis of multiple comparisons. For comparison between treatment groups, the null hypothesis was tested by either single-factor analysis of variance (ANOVA) for multiple groups, or the unpaired *t*-test for two groups, and the data are presented as mean ± SD.

The coefficient of drug interaction (CDI) has been calculated as follows: CDI = AB/(A × B). According to the values of each group, AB is the ratio of the combination groups to control group; A or B is the ratio of the single agent group to control group. Thus, CDI value <1, =1, or >1 indicates that the drugs are synergistic, additive, or antagonistic, respectively.

## Figures and Tables

**Figure 1 ijms-20-02593-f001:**
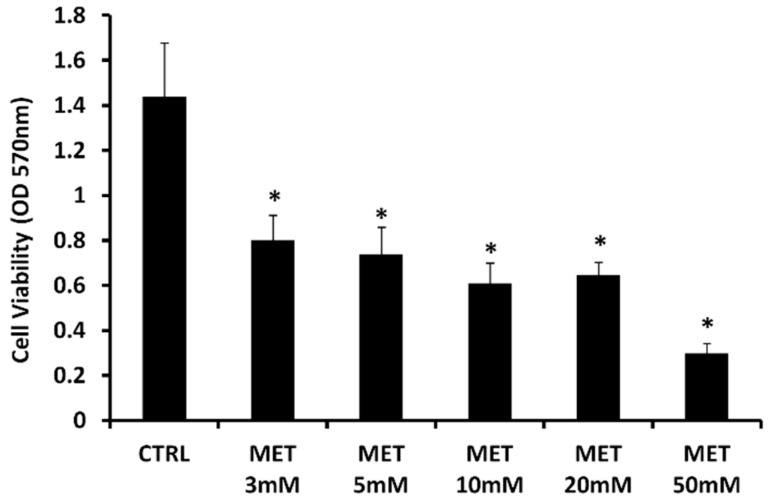
Metformin (MET) caused a decrease in cell viability. Cell viability, determined using MTT assay, of androgen-independent human prostate cancer cells (DU145), control untreated (CTRL) and treated with Metformin at different concentrations (3, 5, 10, 20, and 50 mM) for 24 h. Data are the means ± SD of three experiments performed in triplicate * *p* < 0.05 versus DU145 untreated cells.

**Figure 2 ijms-20-02593-f002:**
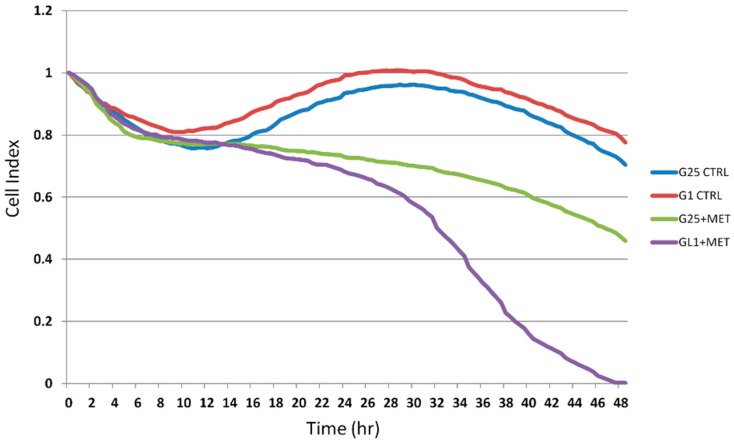
Metformin decrease cell proliferation in presence of different glucose concentrations. DU145 proliferation in the different groups recorded in real time, using the xCELLigence system. The cells showed growth with 1 mM glucose (G1) or 25 mM glucose (G25) in presence and absence of 10 mM metformin (MET).

**Figure 3 ijms-20-02593-f003:**
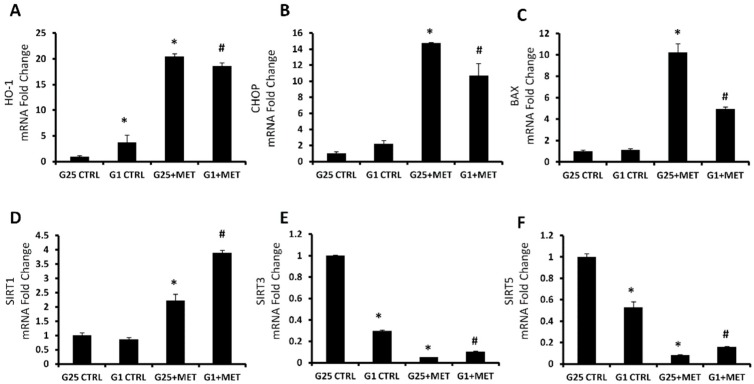
MRNA expression of HO-1 (**A**), CHOP (**B**), BAX (**C**), SIRT1 (**D**), SIRT3 (**E**), and SIRT5 (**F**), of control cells with 25 mM glucose (G25 CTRL), control cells with 1 mM glucose (G1 CTRL), G25 treated with 10 mM metformin (G25 + MET), and G1 treated with 10 mM metformin (G1 + MET). Results are mean ± SD, * *p* < 0.05 vs. G25 CTRL, ^#^
*p* < 0.05 vs. G1 CTRL.

**Figure 4 ijms-20-02593-f004:**
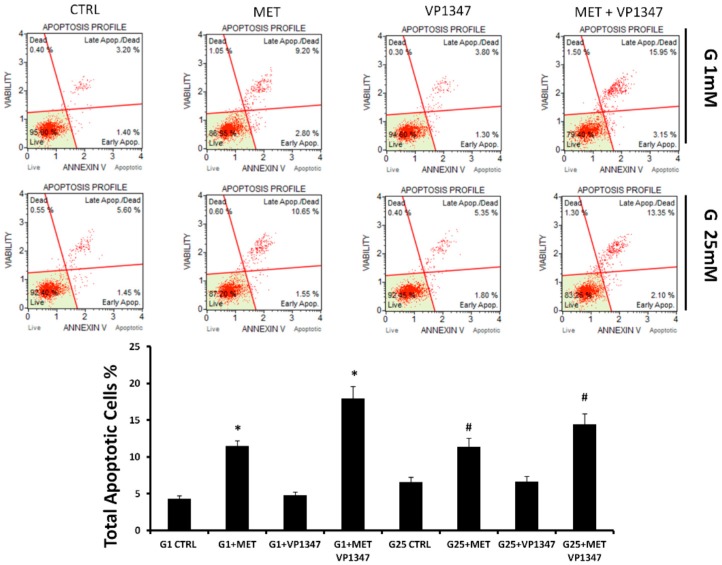
Effect of metformin and HO-1 activity inhibitor VP1347 on DU145 cells apoptosis. Cells were incubated with 1 mM glucose (G1) or 25 mM glucose (G25) in the presence and absence of 10 mM metformin (MET) and 10 μM VP1347 for 12 h. Apoptosis was evaluated by cytometry, using the Muse Annexin V and Dead Cell Assay Kit. The graph showed the total apoptotic cells percentage in the different groups. * *p* < 0.05 vs. G1 CTRL, ^#^
*p* < 0.05 vs. G25 CTRL.

**Figure 5 ijms-20-02593-f005:**
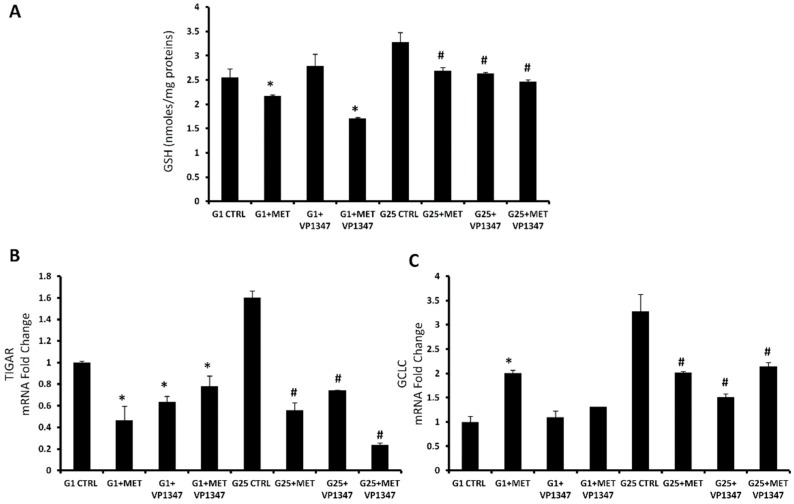
Metformin regulates oxidative stress pathway. (**A**) Thiol groups in DU145 cells treated for 24 h with 10 mM metformin and 10 uM VP1347. Thiol groups are expressed as nmol/mg protein. Values represent the means ± SD of three experiments performed in triplicate. * *p* < 0.05, significant result vs. untreated DU145 cells. (**B**,**C**) MRNA expression of TIGAR and GCLC of control cells with 25 mM glucose (G25), control cells with 1 mM glucose (G1), in the presence and absence of 10 mM metformin (MET) and 10 µM VP1347 for 6 h. Results are mean ± SD, * *p* < 0.05 vs. G25 CTRL, ^#^
*p* < 0.05 vs. G1 CTRL.

**Figure 6 ijms-20-02593-f006:**
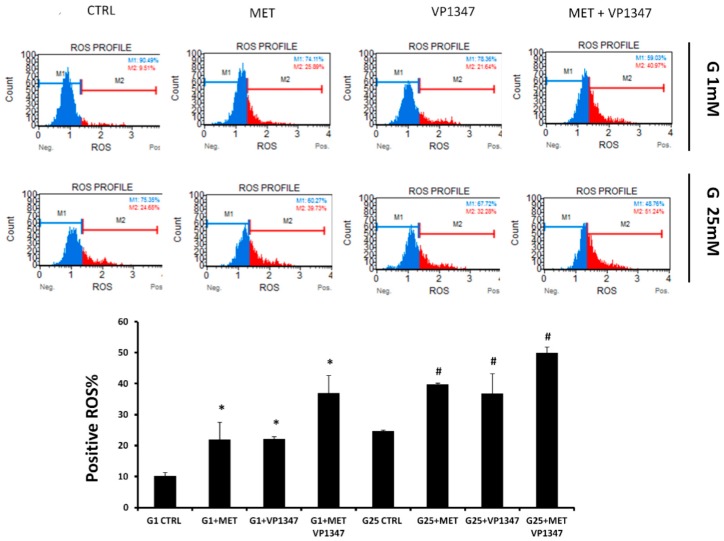
Effect of Metformin and HO-1 activity inhibitor VP1347 on DU145 cells ROS production. Cells were incubated with 1 mM glucose (G1) or 25 mM glucose (G25) in the presence and absence of 10 mM metformin (MET) and 10 µM VP1347 for 6 h. The quantitative measurement of cells undergoing oxidative stress was evaluated by cytometry, using the Muse Oxidative Stress Kit. The graph showed the positive ROS percentage in the different groups. * *p* < 0.05 vs. G1 CTRL, ^#^
*p* < 0.05 vs. G25 CTRL.

**Figure 7 ijms-20-02593-f007:**
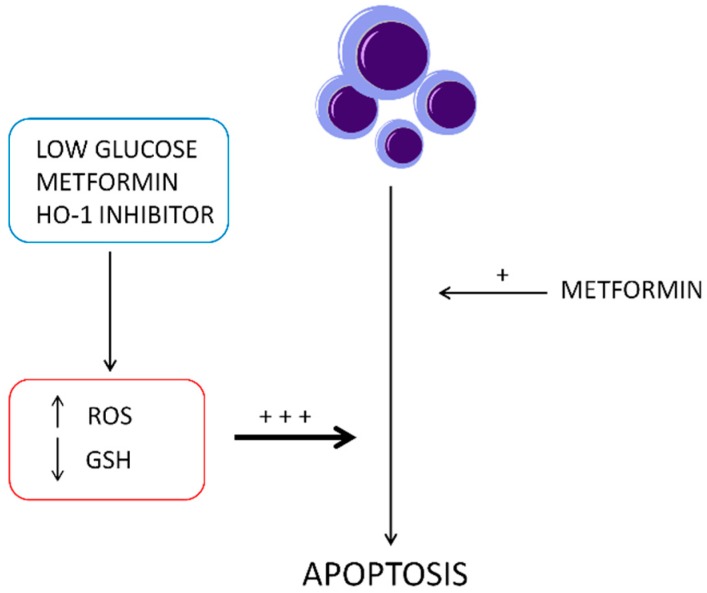
Schematic description of metformin and HO-1 activity inhibitor (VP1347) synergistic effect to induce apoptosis in DU145 cells.

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
