# Peer review of "Heme Oxygenase-1 Inhibition Sensitizes Human Prostate Cancer Cells towards Glucose Deprivation and Metformin-Mediated Cell Death"

_ijms, 2019, doi:10.3390/ijms20102593_

Round 1

Reviewer 1 Report

Significant work by Dr. Vanella and colleagues. Few minor changes should be done before it is ready for acceptance. They are as follows:

Authors should come up with a model depicting the theme of this work and add it to the manuscript as Fig 7. This will help the readers to get an overall idea about the significance of this work. Authors should explain the model in discussion section.

Two major concepts should be discussed in the discussion section. a) Authors should mention about their thought on how glucose starvation might affect cell cycle metabolic check points on basis of their work. For reference purposes they can check this work:  doi: 10.18632/oncoscience.253/ PMCID: PMC4671930/ PMID: 26682255 and b) Author's thought on how other metformin like drugs like AICAR would act in this scenario. As mentioned in doi: 10.1080/15384101.2015.1087623/ PMCID: PMC4825547/ PMID: 26323019.

Specifically how AICAR or phenformin would work similar like metformin been shown worked with HMOX1 inhibitor. They could refer the above mentioned works in case needed.

Few minor changes:

mention the x axis title in fig 2

increase the size of word in figures 3 and 5. 

Author Response

The authors thank the reviewer for the handling of our manuscript. We have responded to the concerns raised about our manuscript. I am happy to say that we have significantly improved the manuscript and responded to all of the reviewer’s concerns.

Comments Reviewer #1

Significant work by Dr. Vanella and colleagues. Few minor changes should be done before it is ready for acceptance. They are as follows:

Authors should come up with a model depicting the theme of this work and add it to the manuscript as Fig 7. This will help the readers to get an overall idea about the significance of this work. Authors should explain the model in discussion section.

An explanatory scheme has been added as Figure 7 and mentioned in the discussion section.

Two major concepts should be discussed in the discussion section. a) Authors should mention about their thought on how glucose starvation might affect cell cycle metabolic check points on basis of their work. For reference purposes they can check this work:  doi: 10.18632/oncoscience.253/ PMCID: PMC4671930/ PMID: 26682255 and b) Author's thought on how other metformin like drugs like AICAR would act in this scenario. As mentioned in doi: 10.1080/15384101.2015.1087623/ PMCID: PMC4825547/ PMID: 26323019.

Specifically how AICAR or phenformin would work similar like metformin been shown worked with HMOX1 inhibitor. They could refer the above mentioned works in case needed.

The discussion has been improved and the suggested references were inserted.

Few minor changes:

mention the x axis title in fig 2

It has been fixed

increase the size of word in figures 3 and 5. 

It has been fixed

Reviewer 2 Report

In this manuscript, the DU145 cancer cell line has been treated with “metformin” and “VP1347” in different glucose levels. Then the proliferation, apoptosis rate and ROS generation of the cells has been measured. Also, they provided information about the expression levels of CHOP, BAX and Sirtuins.

In my review, in vitro studies to evaluate the effects metformin on prostate cancer progression are clinically relevant and further information about synthesizing prostate cancer cells to metformin could be translatable. Regrettably, the presented data in this manuscript are not convincing enough to be considered for publication as an original article in its current format. After the following revisions, the manuscript could be re-submitted for re-review:

1-      Inhibition of HO-1 should be clearly presented and conformed. “VP1347” is referred as a selective inhibitor of HO-1 activity while in the “material and method section” there is not information about either its production by the authors or the name of the company which has been provided it. On the other hand, on Line 224&5 the authors mentioned that they used a HO-1 inhibitor.  

The mechanism of action of “VP1347” should be elaborated. A western blot analysis should be done to show the protein levels of HO-1, a few downstream targets of HO-1 and apoptotic markers before and after inhibition.

2-      DU145 is an androgen receptor (AR)-negative and PTEN positive cell line and it is not clear to me why this single cell line is selected for this experiment. If the objective is to show the proposed combination is appropriate for high grade prostate cancer, then PC3 cell line (AR-/PTEN-) was a better choice. All the experiments should be done using at least one more cell line preferably PC3 (AR-/PTEN-) and LNCaP (AR+/PTEN+).

3-      The drug treatment data should be re-analyzed to see if the observed effect is additive or synergistic.

Author Response

The authors thank the reviewer for the handling of our manuscript. We have responded to the concerns raised about our manuscript. I am happy to say that we have significantly improved the manuscript and responded to all of the reviewer’s concerns.

Comments Reviewer #2

In this manuscript, the DU145 cancer cell line has been treated with “metformin” and “VP1347” in different glucose levels. Then the proliferation, apoptosis rate and ROS generation of the cells has been measured. Also, they provided information about the expression levels of CHOP, BAX and Sirtuins.

In my review, in vitro studies to evaluate the effects metformin on prostate cancer progression are clinically relevant and further information about synthesizing prostate cancer cells to metformin could be translatable. Regrettably, the presented data in this manuscript are not convincing enough to be considered for publication as an original article in its current format. After the following revisions, the manuscript could be re-submitted for re-review:

 1-      Inhibition of HO-1 should be clearly presented and conformed. “VP1347” is referred as a selective inhibitor of HO-1 activity while in the “material and method section” there is not information about either its production by the authors or the name of the company which has been provided it. On the other hand, on Line 224&5 the authors mentioned that they used a HO-1 inhibitor.  The mechanism of action of “VP1347” should be elaborated. A western blot analysis should be done to show the protein levels of HO-1, a few downstream targets of HO-1 and apoptotic markers before and after inhibition.

The statement “HO-1 activity inhibitor” is now conform in the text.

“Cells were treated and incubated with metformin 10mM and VP1347, 1-{4-[(4-bromobenzyl)oxy]phenyl}-2-(1H-imidazol-1-yl)ethanol, a HO-1 activity inhibitor, synthesized by Salerno et al. from University of Catania as previously described (Salerno et al. "Potholing of the Hydrophobic Heme Oxygenase-1 Western Region for the Search of Potent and Selective Imidazole-Based Inhibitors." Eur J Med Chem 148 (2018): 54-62.)”.

As shown in the just mentioned manuscript, VP1347 has been synthesized as a specific inhibitor of Heme Oxygenase-1 enzymatic activity. Our preliminary results show that HO-1 protein expression is not affected by VP1347 treatment. After western blot and densitometric analysis, using Beta-Actin as housekeeping protein, the levels of protein were 0.640 for Control and 0.676 for VP1347 group under low glucose condition. Under high glucose condition the levels of protein were 0.419 for Control and 0.504 for VP1347 group (data not shown).

The new synthesized compound, VP1347, is a non competitive inhibitor and binds to the complex enzyme/substrate.

The aim of this preliminary study is to show for the first time the role of Heme Oxygenase-1 on metformin mediated effects in metastatic prostate cancer under low and high glucose conditions.

We schedule to perform the additional protein analysis and the complete molecular mechanism after a new refunding of the project.

2-      DU145 is an androgen receptor (AR)-negative and PTEN positive cell line and it is not clear to me why this single cell line is selected for this experiment. If the objective is to show the proposed combination is appropriate for high grade prostate cancer, then PC3 cell line (AR-/PTEN-) was a better choice. All the experiments should be done using at least one more cell line preferably PC3 (AR-/PTEN-) and LNCaP (AR+/PTEN+).

The evolution from local PCa to castration-resistant PCa, an end-stage of disease, is often associated with changes in genes such as p53, androgen receptor, PTEN, and ETS gene fusion products (Tran et al., Combination Therapies Using Metformin and/or Valproic Acid in Prostate Cancer: Possible Mechanistic Interactions. Curr Cancer Drug Targets. 2018 Jul 23). HO-1 was found increased under the malignant condition, as previously demonstrated by Maines et al. (Expression of heme oxygenase-1 (HSP32) in human prostate: normal, hyperplastic, and tumor tissue distribution. Urology. 1996 May;47(5):727-33). For this study, we decided to use the DU145 cell line because, as showed in our previous manuscript (“Antiproliferative effect of oleuropein in prostate cell lines. Acquaviva et al., Int J Oncol. 2012 Jul;41(1):31-8. doi: 10.3892/ijo.2012.1428. Epub 2012 Apr 5”), the DU145 cells have a higher basal expression of HO-1 compared to the other cell lines (BPH-1 and LNCaP). In particular, the levels of Heme Oxygenase-1, measured by ELISA assay, were 3.215 ng/mg proteins as concern LNCaP control cells and 14.461 ng/mg proteins as concern DU145 control cells. For this reason, we supposed that DU145 cells would have been more susceptible towards the inhibition of HO-1 activity.

3-      The drug treatment data should be re-analyzed to see if the observed effect is additive or synergistic.

Based on ROS and Annexin V data (Figures 4 and 6), the VP1347 shows slight effects when used alone, while as co-treatment it increases the metformin effects, indicating a synergistic action between the two drugs. 

Reviewer 3 Report

This manuscript provides information which indicate HO-1 inhibition in combination with Metformin treatment can inhibit the proliferation of prostate cancer cells and promote prostate cancer cell apoptosis. The studies are conducted using a single prostate cancer cell line, DU145. This is a major weakness; the impact of treatment on at least one more cell line needs to be performed in order for the results to have meaning. The experimental design and data analyses are compelling and well described. Inclusion of analyses using the xCELLigence system and oxidative stress assays are a strength. There are some grammatical errors and incorrect word usage throughout the manuscript that are somewhat distracting but can be easily fixed. Other concerns are listed below;

1.       Introduction and discussion: There are too many short paragraphs (some only one line in length) throughout the Introduction and Discussion sections –these need to be combined and/or expanded on to make the Introduction and Discussion sections less ‘choppy’ and easier to read

2.       Introduction: please note which cell types typically express HO-1 in a healthy patient and provide specific details of the molecular mechanisms which can cause dysregulation of HO-1 expression can occur

3.       Lines 39 – 40: This one sentence paragraph needs to be expanded on; please provide some details of the mechanisms by which high levels of HO-1 contribute to drug resistance and regulation of cell redox homeostasis. Please also note which cancer types have been associated with high levels of HO-1

4.       Introduction: please state the mechanism of action of Metformin in regards to how it affects glucose metabolism. Please also state the mechanism of action of simvastatin/atorvastatin as well as valproic acid (only a few words are needed, but this is important to provide context)

5.       Lines 64 – 66: The authors state ‘to the best of our knowledge, to date, no studies have specifically focused on simultaneously glucose deprivation, anti-proliferative properties of metformin and its precise molecular mechanism in prostate cancer’. A PubMed search using the search terms ‘prostate cancer’ and ‘metformin’ generated 261 results, and many of these papers discuss the role of Metformin-mediated glucose deprivation in reducing prostate cancer cell proliferation and provide insights into how this occurs. Data from these papers should be summarized in the manuscript and citations included. The authors may want to consider including this information in the Discussion section rather than the Introduction section, but again my major concern is that I feel their statement in lines 64 – 66 is misleading.

6.       The impact of HO-1 inhibition (and HO-1 plus/minus metformin treatment) needs to be assessed in more than one cell line; this is a major weakness

7.       Lines 78 – 79, and line 244: please note whether or not physiologically acceptable doses of metformin are used in the study (i.e. doses that can be safely administered to patients)  

8.       Line 283: the acronym ‘RSH’ needs to be explained

9.       Line 304: please explain why SEM values and not SD values are reported, it seems that SD values would be more appropriate

10.   Figure legends: It would be helpful to briefly comment on the key finding(s) of the experiment for each figure legend. E.g. For figure one the authors could note that treatment with metformin causes a dose-dependent decrease in cell viability

11.   Figure 2: the x-axis needs to be labeled (I assume this is time), and acronyms, e.g. MET, need to be explained

12.   Figure 3: It would be helpful to perform western blot analyses as well as qPCR to determine the impact of treatment on protein expression levels of these molecules – this would allow for a better understanding of which molecules play a key role in mediating the physiological changes that are observed

13.   Introduction: The relationship between metformin and SIRTs needs to be explained in the introduction so that the reader can understand the relevance of assessing SIRT expression. Likewise for the other molecules being assessed, including TIGAR (this acronym isn’t explained until the discussion section (line 217) – it needs to be introduced earlier in the manuscript). The acronym GSH also needs to be explained and put in context.

14.   Discussion: DU145 cells harbor mutant p53; please comment on how this could impact response to the treatments. Again, including additional cell lines is important, I would advise the authors to select cell lines with wildtype p53 to determine whether p53 status impacts response

15.   Discussion: Metformin reduces systemic glucose levels. This isn’t discussed anywhere in the manuscript; the authors should put this information in the manuscript/in context of the findings

16.   Lines 228 – 229: There should be some mention of HO-1 in this final sentence/paragraph; HO-1 is a major focus of the manuscript

Author Response

The authors thank the reviewer for the handling of our manuscript. We have responded to the concerns raised about our manuscript. I am happy to say that we have significantly improved the manuscript and responded to all of the reviewer’s concerns.

Comments Reviewer #3

This manuscript provides information which indicate HO-1 inhibition in combination with Metformin treatment can inhibit the proliferation of prostate cancer cells and promote prostate cancer cell apoptosis. The studies are conducted using a single prostate cancer cell line, DU145. This is a major weakness; the impact of treatment on at least one more cell line needs to be performed in order for the results to have meaning. The experimental design and data analyses are compelling and well described. Inclusion of analyses using the xCELLigence system and oxidative stress assays are a strength. There are some grammatical errors and incorrect word usage throughout the manuscript that are somewhat distracting but can be easily fixed. Other concerns are listed below;

1.       Introduction and discussion: There are too many short paragraphs (some only one line in length) throughout the Introduction and Discussion sections –these need to be combined and/or expanded on to make the Introduction and Discussion sections less ‘choppy’ and easier to read

It has been improved as suggested.

2.       Introduction: please note which cell types typically express HO-1 in a healthy patient and provide specific details of the molecular mechanisms which can cause dysregulation of HO-1 expression can occur

Heme Oxygenase-1, as an inducible form, is low expressed in healthy tissues but protein expression can be induced by several stimuli including reactive oxygen species. It’s well known that many cancers present high levels of oxidative stress and hyper-activated antioxidant defenses.

3.       Lines 39 – 40: This one sentence paragraph needs to be expanded on; please provide some details of the mechanisms by which high levels of HO-1 contribute to drug resistance and regulation of cell redox homeostasis. Please also note which cancer types have been associated with high levels of HO-1.

High levels of Heme Oxygenase-1 have been shown in many cancers as reported by Jozkowicz et al. (Heme Oxygenase-1 in Tumors. Antioxid Redox Signal. 2007 Dec; 9(12): 2099–2117). A previous study by Florczyk et al. (Overexpression of biliverdin reductase enhances resistance to chemotherapeutics.  Cancer Lett. 2011 Jan 1;300(1):40-7. Epub 2010 Oct 8.), revealed that enhanced activity of biliverdin reductase may protect cells in stressful conditions arising from anti-cancer drugs, cisplatin and doxorubicin.  Additionally, data from Banerjee et al. (Heme Oxygenase-1 Promotes Survival of Renal Cancer Cells through Modulation of Apoptosis- and Autophagy-regulating Molecules.  September 14, 2012, The Journal of Biological Chemistry 287, 32113-32123) demonstrate that HO-1 is up-regulated in renal cancer cells as a survival strategy against chemotherapeutic drugs and promotes growth of tumor cells by inhibiting both apoptosis and autophagy. Thus, application of chemotherapeutic drugs along with HO-1 inhibitor may elevate therapeutic efficiency by reducing the cytoprotective effects of HO-1 and by simultaneous induction of both apoptosis and autophagy.

4.       Introduction: please state the mechanism of action of Metformin in regards to how it affects glucose metabolism. Please also state the mechanism of action of simvastatin/atorvastatin as well as valproic acid (only a few words are needed, but this is important to provide context)

Metformin treatment and caloric restriction increase the AMP:ATP ratio and activate AMPK, switching cells from an anabolic to a catabolic state. Treatment of breast cancer cells with metformin significantly decreased cholesterol content with concomitant inhibition of various cholesterol regulatory genes (e.g., HMGCoR, LDLR and SREBP1) (Sharma et al., Metformin exhibited anticancer activity by lowering cellular cholesterol content in breast cancer cells. PLoS One. 2019 Jan 9;14(1):e0209435. eCollection 2019), suggesting that using drugs affecting cholesterol synthesis, such as simvastatin/atorvastatin, could be used in the treatment of cancer (Hindler et al.,The Role of Statins in Cancer Therapy. The Oncologist March 2006 vol. 11 no. 3 306-315).

5.       Lines 64 – 66: The authors state ‘to the best of our knowledge, to date, no studies have specifically focused on simultaneously glucose deprivation, anti-proliferative properties of metformin and its precise molecular mechanism in prostate cancer’. A PubMed search using the search terms ‘prostate cancer’ and ‘metformin’ generated 261 results, and many of these papers discuss the role of Metformin-mediated glucose deprivation in reducing prostate cancer cell proliferation and provide insights into how this occurs. Data from these papers should be summarized in the manuscript and citations included. The authors may want to consider including this information in the Discussion section rather than the Introduction section, but again my major concern is that I feel their statement in lines 64 – 66 is misleading.

The statement was deleted as suggested and new references were added.

6.       The impact of HO-1 inhibition (and HO-1 plus/minus metformin treatment) needs to be assessed in more than one cell line; this is a major weakness

The evolution from local PCa to castration-resistant PCa, an end-stage of disease, is often associated with changes in genes such as p53, androgen receptor, PTEN, and ETS gene fusion products (Tran et al., Combination Therapies Using Metformin and/or Valproic Acid in Prostate Cancer: Possible Mechanistic Interactions. Curr Cancer Drug Targets. 2018 Jul 23). HO-1 was found increased under the malignant condition, as previously demonstrated by Maines et al. (Expression of heme oxygenase-1 (HSP32) in human prostate: normal, hyperplastic, and tumor tissue distribution. Urology. 1996 May;47(5):727-33). For this study, we decided to use the DU145 cell line because, as showed in our previous manuscript (“Antiproliferative effect of oleuropein in prostate cell lines. Acquaviva et al., Int J Oncol. 2012 Jul;41(1):31-8. doi: 10.3892/ijo.2012.1428. Epub 2012 Apr 5”), the DU145 cells have a higher basal expression of HO-1 compared to the other cell lines (BPH-1 and LNCaP). In particular, the levels of Heme Oxygenase-1, measured by ELISA assay, were 3.215 ng/mg proteins as concern LNCaP control cells and 14.461 ng/mg proteins as concern DU145 control cells. For this reason, we supposed that DU145 cells would have been more susceptible towards the inhibition of HO-1 activity. The aim of this preliminary study is to show for the first time the role of Heme Oxygenase-1 on metformin mediated effects in metastatic prostate cancer under low and high glucose conditions.

We schedule to perform the additional analysis and the complete molecular mechanism in different cell lines after a new refunding of the project.

7.       Lines 78 – 79, and line 244: please note whether or not physiologically acceptable doses of metformin are used in the study (i.e. doses that can be safely administered to patients)  

Several studies used supra-physiological (0.5-50mM) and physiological concentrations (<0.5mM) of metformin to investigate its anti-apoptotic effects. High concentration of metformin are used as a proof of concept to study the molecular pathway.

8.       Line 283: the acronym ‘RSH’ needs to be explained

It has been fixed.

9.       Line 304: please explain why SEM values and not SD values are reported, it seems that SD values would be more appropriate

We apologize for the mistake in the text, but all SEM values were SD value. They are now conformed.

10.   Figure legends: It would be helpful to briefly comment on the key finding(s) of the experiment for each figure legend. E.g. For figure one the authors could note that treatment with metformin causes a dose-dependent decrease in cell viability.

Figure legends has been improved as suggested.

11.   Figure 2: the x-axis needs to be labeled (I assume this is time), and acronyms, e.g. MET, need to be explained

It has been fixed.

12.   Figure 3: It would be helpful to perform western blot analyses as well as qPCR to determine the impact of treatment on protein expression levels of these molecules – this would allow for a better understanding of which molecules play a key role in mediating the physiological changes that are observed

The aim of this preliminary study is to show for the first time the role of Heme Oxygenase-1 on metformin mediated effects in metastatic prostate cancer under low and high glucose conditions.

Our preliminary results show that HO-1 protein expression is not affected by VP1347 treatment. After western blot and densitometric analysis, using Beta-Actin as housekeeping protein, the levels of protein were 0.640 for Control and 0.676 for VP1347 group under low glucose condition. Under high glucose condition the levels of protein were 0.419 for Control and 0.504 for VP1347 group (data not shown).

We schedule to perform the additional protein analysis and the complete molecular mechanism after a new refunding of the project.

13.   Introduction: The relationship between metformin and SIRTs needs to be explained in the introduction so that the reader can understand the relevance of assessing SIRT expression. Likewise for the other molecules being assessed, including TIGAR (this acronym isn’t explained until the discussion section (line 217) – it needs to be introduced earlier in the manuscript). The acronym GSH also needs to be explained and put in context.

Introduction has been improved as followed:

 Pharmacological AMPK activation led to SIRT1 activation. Metformin (an AMPK activator) has been proposed to operate as an agonist of SIRT1, a nicotinamide adenine dinucleotide (NAD+)-dependent deacetylase that mimics most of the metabolic responses to calorie restriction (Cuyàs et al., Metformin Is a Direct SIRT1-Activating Compound: Computational Modeling and Experimental Validation. Front. Endocrinol., 06 November 2018).

Metformin significantly decreased the intracellular glutathione levels and enhanced sensitivity of esophageal squamous cell carcinoma to cisplatin in vitro and in vivo, suggesting that regulation of the antioxidant defenses represent a key target for cancer therapy. To this regard new evidences have shown the involvement of TIGAR (TP53-induced glycolysis and apoptosis regulator) in glutathione restoration.

14.   Discussion: DU145 cells harbor mutant p53; please comment on how this could impact response to the treatments. Again, including additional cell lines is important, I would advise the authors to select cell lines with wildtype p53 to determine whether p53 status impacts response

It has been shown that metformin is selectively toxic to p53-deficient cells and provides a potential mechanism for the reduced incidence of tumors observed in patients being treated with metformin (Buzzai et al., Systemic treatment with the antidiabetic drug metformin selectively impairs p53-deficient tumor cell growth. Cancer Res. 2007 Jul 15;67(14):6745-52). We schedule to perform the additional analysis and the complete molecular mechanism in different cell lines after a new refunding of the project.

15.   Discussion: Metformin reduces systemic glucose levels. This isn’t discussed anywhere in the manuscript; the authors should put this information in the manuscript/in context of the findings

In diabetic patients, metformin decreases plasma glucose concentration mainly by decreasing hepatic gluconeogenesis and glucose output. This effect is followed by an increase in glucose uptake and the amelioration of insulin resistance (Salani et al., Metformin, cancer and glucose metabolism, Endocr Relat Cancer. 2014;21(6):R461-71).

Metformin works by targeting the enzyme AMPK (AMP activated protein kinase), which induces muscles to take up glucose from the blood. A recent breakthrough has found the upstream regulator of AMPK to be a protein kinase known as LKB1, a well-recognized tumor suppressor (Evans et al., Metformin and reduced risk of cancer in diabetic patients, BMJ. 2005 Jun 4;330(7503):1304-5).

16.   Lines 228 – 229: There should be some mention of HO-1 in this final sentence/paragraph; HO-1 is a major focus of the manuscript

It has been improved as suggested.

Round 2

Reviewer 2 Report

The new version of this manuscript could be considered for publication on IJMS after a minor revision as a “Communication” rather than an “Article”. 

As the authors mentioned in their replies, this manuscript contains preliminary data, and more investigations are needed to characterize the apparent inhibition of HO-1 activity by this non-competitive inhibitor of enzyme/substrate. 

To me, the research design is still poor, and if the objective was the evaluation of HO-1 inhibition on metformin efficiency, a well-characterized inhibitor of HO-1 could be used as a positive control. 

After following minor revision, the manuscript should be re-evaluated: 

1-         The labels in horizontal (treatment) axis should be enhanced. CTRL should be replaced with either Control or DMSO or,… 

Any abbreviation should be defined when you use it for the first time (G, MET, …)

2-         Fig. 2 contains redundant time information (hh:mm:ss). Add markers to the lines and put a reasonable scale for the time.    

3-         Report the coefficient of drug interaction

Author Response

The new version of this manuscript could be considered for publication on IJMS after a minor revision as a “Communication” rather than an “Article”. 

As the authors mentioned in their replies, this manuscript contains preliminary data, and more investigations are needed to characterize the apparent inhibition of HO-1 activity by this non-competitive inhibitor of enzyme/substrate. 

To me, the research design is still poor, and if the objective was the evaluation of HO-1 inhibition on metformin efficiency, a well-characterized inhibitor of HO-1 could be used as a positive control. 

All the well-characterized inhibitor of heme oxygenase, such as SnMP or ZnPP, are not selective because they inhibit both HO-1 and HO-2 activity. Notably, these drugs even increase the protein expression of HO-1. Our new synthesized compound is selective for the HO-1 isoform and inhibits only the activity of HO-1, which represents the inducible form involved in chemoresistance. The aim of this study was to reverse drug-resistance of DU145 observed in vitro using a specific activity HO-1 inhibitor.

Abate et al., “The role of Bach1 in the induction of heme oxygenase by tin mesoporphyrin”. Biochem Biophys Res Commun. 2007 Mar 16;354(3):757-63.

Sardana et al., “Dual control mechanism for heme oxygenase: tin(IV)-protoporphyrin potently inhibits enzyme activity while markedly increasing content of enzyme protein in liver”. Proc Natl Acad Sci U S A. 1987 Apr;84(8):2464-8.

After following minor revision, the manuscript should be re-evaluated:

1-    The labels in horizontal (treatment) axis should be enhanced. CTRL should be replaced with either Control or DMSO or,… 

Any abbreviation should be defined when you use it for the first time (G, MET, …)

All the abbreviations are now defined in the figures legend and in the text as suggested.

2-    Fig. 2 contains redundant time information (hh:mm:ss). Add markers to the lines and put a reasonable scale for the time. 

It has been improved as suggested.

3-    Report the coefficient of drug interaction

We reported the CDI values in the text as suggested.

The Coefficient of drug interaction (CDI) has been calculated as follows: CDI = AB/(A x B). According to the values of each group, AB is the ratio of the combination groups to control group; A or B is the ratio of the single agent group to control group. Thus, CDI value <1, = 1 or >1 indicates that the drugs are synergistic, additive or antagonistic, respectively.

Reviewer 3 Report

The authors have adequately addressed the majority of my concerns - thank you. The authors state that they will conduct future studies using additional cell lines, however, I feel strongly that at least some of the experiments that are part of the current paper need to be repeated using other cell lines.

Author Response

The authors have adequately addressed the majority of my concerns - thank you. The authors state that they will conduct future studies using additional cell lines, however, I feel strongly that at least some of the experiments that are part of the current paper need to be repeated using other cell lines.

DU145 cells represent an in vitro model of chemo-resistant cancer cells but regrettably not so much studied compared to other cells lines. Additionally, even metformin is less active on DU145 proliferation. As reported by Sahra et al., metformin was significantly more powerful to inhibit cell proliferation in LNCaP (52% decrease) than in DU145 (26%) and PC3 (22%), p53 mutated and null cells respectively, highlighting the important role of p53 in the anti-proliferative effect of metformin in prostate cancer cells (Metformin, independent of AMPK, induces mTOR inhibition and cell cycle arrest through REDD1; PMID: 21540236).

Results obtained by Gonnissen et al., showed that p53-mutant cells were more resistant to MET than PC3 and 22Rv1 (PMID: 28208838).

In order to determine whether MET affects proliferation of human p53-mutant prostate cancer cells, we analyzed the effect of the drug on DU145, a p53-mutant cell line.

The aim of this study was to reverse drug-resistance of DU145 observed in vitro using a specific activity HO-1 inhibitor.

Round 3

Reviewer 3 Report

There are several prostate cancer cell lines that harbor p53 mutations in addition to Du145; e.g. LAPC-4, PC-3, VCaP. The authors should repeat some of the experiments using additional prostate cancer cell lines to ensure that the data reported are reproducible.

Author Response

We agree with the Editor, and we thank him and the reviewers for the opportunity to publish our innovative preliminary results as Communication.